# Effectiveness of intravenous albumin therapy to prevent spontaneous bacterial peritonitis, renal dysfunction and death in adults with cirrhosis: a protocol for a systematic review

Suvi Härmälä,[1] Constantinos Parisinos,[1] Jennifer Ryan,[2] Alastair O'Brien[3]

[1]Institute of Health Informatics, University College London, London, UK
[2]Royal Free London NHS Foundation Trust, London, UK
[3]Division of Medicine, University College London, London, UK

**Correspondence to**
Suvi Härmälä;
suvi.harmala.14@ucl.ac.uk

## ABSTRACT

**Introduction** Use of albumin therapy is recommended for management of disease complications in cirrhosis. The effectiveness of albumin to prevent specific disease complications and death, however, is less clear.

**Methods and analysis** We will search Medline (Ovid), Embase (Ovid), Cochrane Hepato-Biliary Controlled Trials Register and Cochrane Central Register of Controlled Trials for published reports on randomised controlled trials and observational studies on the effectiveness of intravenous albumin therapy to prevent spontaneous bacterial peritonitis, renal dysfunction and death in cirrhotic patients. Two independent reviewers will screen the studies for eligibility, extract data and assess risk of bias and quality of evidence using Grading of Recommendations Assessment, Development and Evaluation system. Random effects meta-analyses will be performed when appropriate.

**Ethics and dissemination** As no primary data will be collected, a formal ethical approval is not required. We plan to publish the results of this study in a relevant peer-reviewed journal or journals. The study results may also be presented at relevant conferences and meetings.

**PROSPERO registration number** CRD42018100798.

### Strengths and limitations of this study

► The broad range of outcomes included this review provide clinical practice and future guidelines a comprehensive picture of the effects of albumin therapy in cirrhosis.

► This study protocol has been developed according to the recommendations of the Preferred Reporting Items for Systematic Reviews and Meta-Analyses Protocols.

► The selection of studies, data extraction, the risk of bias and quality of evidence assessment using Grading of Recommendations Assessment, Development and Evaluation system will be conducted by two independent authors.

► The eligibility criteria used may not result in the selection of studies that are homogeneous in methods limiting the ability to draw reliable conclusions.

► Inclusion of studies regardless of the type of clinical setting or frequency of albumin delivery may limit the practical applicability of the summarised therapy effects to all clinical settings.

## BACKGROUND

More than 45 million people globally have cirrhosis and other severe forms of chronic liver disease.[1] Driven by upward trend in obesity and alcohol consumption, the prevalence of liver disease is increasing. Chronic liver disease, due to fibrosis-causing inflammation, usually progresses silently and slowly until the functioning of the liver is severely compromised and patients develop life-threatening complications such as ascites, spontaneous bacterial peritonitis (SBP) and renal dysfunction.

Albumin, the most abundant protein in human serum, has an important role in both maintaining fluid distribution in the body and potentially regulating immune response by binding and inactivating pro-inflammatory molecules. In advanced liver disease, the synthesis of albumin in the liver is disturbed and both the quantity and functionality of albumin are substantially reduced.[2] Treating patients with advanced liver disease with albumin infusions might improve both their ability to respond to infectious threats such as SBP and ability to restore adequate renal blood flow. Current clinical guidelines for albumin use in decompensated cirrhosis recommend the use of intravenous albumin infusions for management of ascites-related symptoms and paracentesis (removal of ascitic fluid) and for the management of SBP, renal dysfunction and variceal bleeding.[3] Routine albumin use is not recommended for the management of non-SBP infections.[3]

## Rationale for the review

To avoid the potentially devastating consequences of the complications of cirrhosis, a better understanding of the effectiveness of available prevention and treatment strategies would be useful. While the use of albumin in the management of cirrhosis complications is currently recommended and widely employed, the effectiveness of albumin to prevent specific disease complications is less clear. Previous reviews have evaluated the effects of albumin therapy both in cirrhotic patients with infections (SBP and non-SBP)[4 5] and in patients with cirrhosis-related ascites undergoing paracentesis.[6 7] The reviews of albumin use in patients with infections,[4 5] however, were written more than 5 years ago and new studies may have been conducted since. The most recent reviews in albumin use in paracentesis, published in 2012[7] and 2017,[6] on the other hand, have come to contradictory conclusions on the albumin's effectiveness to prevent of death after the procedure. None of the reviews included SBP as a study outcome and non-randomised studies were excluded.

The aim of this review is to improve our understanding of the effects of albumin use in cirrhosis by reviewing the currently available evidence and quantifying the effectiveness of intravenous albumin therapy to prevent specific cirrhosis complications, SBP and renal dysfunction, and death. In contrast to previous reviews, we will also consider evidence from non-randomised studies. The results of this review may be used to inform future guidelines and clinical management of decompensated cirrhosis.

## OBJECTIVES

The aim of this review is to assess the effectiveness of intravenous albumin therapy to prevent SBP, renal dysfunction and death in adults with cirrhosis.

The objectives are:
- ► To assess the effectiveness of intravenous albumin therapy to prevent SBP in adults with cirrhosis and ascites (without SBP or non-SBP infection).
- ► To assess the effectiveness of intravenous albumin therapy to prevent renal dysfunction in adults with cirrhosis and ascites and/or infection (SBP or non-SBP infection).
- ► To assess the effectiveness of intravenous albumin therapy to prevent death in adults with cirrhosis and ascites and/or infection (SBP or non-SBP infection).

## METHODS

This study protocol has been developed following the recommendations of the Preferred Reporting Items for Systematic Reviews and Meta-Analyses Protocols 2015.[8]

### Eligibility criteria
#### Types of studies

We will include randomised clinical trials (RCTs) and cohort (with comparison group/s) and case-control studies that investigate the effectiveness of intravenous albumin therapy in patients with cirrhosis and ascites and/or infection (SBP or non-SBP infection). We will include studies that have been published or accepted for publication in abstract form or in full. We will exclude review articles, meta-analyses, case reports, cross-sectional studies, animal studies, editorials, surveys of medical practice, clinical guidelines and studies that have been retracted from publication.

### Types of participants

We will include studies that enrol more than 18-year-old adult patients with cirrhosis (regardless of severity or aetiology by any classification) and ascites and/or infection (SBP or non-SBP infection), and more than 18-year-old adult patients with cirrhosis and ascites without obvious signs of baseline bacterial infection when an infection (SBP or non-SBP infection) is a study outcome.

### Types of interventions

We will include studies that investigate the effects of intravenously administered albumin in any setting, of any dose, administration frequency and duration of therapy.

### Types of comparators

We will include studies comparing albumin therapy to a placebo, an alternative intervention or no intervention.

### Types of outcome measures

We will include studies that report on one or more of our primary outcomes and/or our secondary outcomes.

#### Primary outcomes
- ► SBP.
- ► Renal dysfunction (hepatorenal syndrome and other forms of renal dysfunction).
- ► All-cause mortality.

#### Secondary outcomes
- ► Non-SBP infections.
- ► Admission to intensive care.
- ► Adverse events (serious adverse events include any adverse event that at any dose results in death, is life-threatening, requires hospitalisation or prolongs existing hospitalisation, results in persistent or significant disability or incapacity, is a congenital anomaly or birth defect, or seriously jeopardises the participant by requiring intervention to prevent one of the above events. All other adverse events will be considered non-serious).

### Information sources
#### Electronic searches

To capture all relevant studies, we plan to search the following databases:
- ► The Cochrane Hepato-Biliary Controlled Trials Register.
- ► Cochrane Central Register of Controlled Trials.
- ► MEDLINE (Ovid).
- ► EMBASE (Ovid).

**Table 1** Medline (Ovid) provisional search terms

| Search concept | Search terms |
|---|---|
| Albumin | 1. Serum Albumin, Human/ |
| | 2. Albumins/ad, tu, th [Administration & Dosage, Therapeutic Use, Therapy] |
| | 3. Albumin.mp. |
| | 4. 1 or 2 or 3 |
| SBP | 5. ('spontaneous bacterial peritonitis' or SBP).mp. |
| Ascites | 6. exp ASCITES/ |
| | 7. ascit*.mp. |
| | 8. 5 or 6 or 7 |
| Cirrhosis | 9. exp Liver Cirrhosis/ |
| | 10. cirrho*.mp. |
| | 11. 9 or 10 |
| Combined search | 12. 4 and 8 and 11 |

Each database will be searched separately for studies published until the date of the search and the search strategy first developed in MEDLINE will be adapted to each database interface as appropriate. Where only a study protocol of an eligible study is found in the search, we will further search for the published study results and if necessary, contact the investigators named in the protocol. We plan to also search relevant studies from the reference lists of the eligible studies identified through the electronic searches and from the previous clinical guidelines for management of patients with cirrhosis.

### Search strategy

We will identify relevant articles by combining search terms for albumin and the eligible base conditions of the study participants: cirrhosis and ascites and/or infection (the search terms for SBP will capture both studies that specify the participants as having 'SBP', 'non-SBP infections' and 'infections other than SBP'). The provisional search terms are listed in table 1. We will not use filters to limit the search.

### Study records
#### Data management
Duplicate records of the same report will be removed using a reference management software (Mendeley). Report screening (both by title only and based on abstract and title), full-text review and extraction of data will be performed using a web-based, systematic review management software (DistillerSR) with standardised online forms. Prior to the review, the forms will be piloted and revised if necessary.

#### Selection process
Articles identified through the search will be first screened by one reviewer (SH) by title only and then by two independent reviewers (SH and CP) by abstract and title.

Records with uncertain eligibility, or subject to disagreement over eligibility, will always be included in the next screening stage until they reach the full-text review.

The full-text review will be completed by two independent review authors (SH and CP). Disagreements over eligibility at this stage will be resolved by discussion and if required by consulting a third review author (AO or JR). Any uncertainties will be resolved by contacting the study investigators. Multiple, overlapping, or companion study reports representing the same study will be combined. If this is not possible, only the report that most closely fulfils our eligibility criteria will be included. The study selection process and reasons for excluding ineligible studies will be recorded and presented in a flow diagram.[9]

### Data collection process
The data will be extracted independently and in duplicate by two review authors (SH and CP). Disagreements will be resolved by discussion and if required by consulting a third review author (AO or JR). Uncertainties will be resolved by contacting the study investigators.

### Data items
The data will be extracted on:
► Study participants: inclusion and exclusion criteria, recruitment/selection method and distribution of baseline characteristics (sex, age, aetiology, severity and characteristics of liver disease, comorbidities, abstinence, medication/treatment other than intervention).
► Interventions and comparison treatments: dose and frequency, size of intervention and comparison groups, length of follow-up.
► Outcomes: definition, time points, number of events, units of measurement, unadjusted and adjusted effect estimates, covariates used for adjustment, quantity of missing data and reasons for missingness, statistical methods.
► Study design: type of study, country, setting, year/s and duration of study.
► Study quality and study bias (as per the assessments specified below).
► Funding and competing interests.

Data on outcome measures will be extracted as reported and, if appropriate, transformed for presentation and analysis.

### Outcomes and prioritisation
Our primary non-fatal outcomes, SBP and renal dysfunction, are common and serious cirrhosis complications. Cirrhotic patients may also experience other disease complications but given albumin is most commonly used clinically to improve blood volume and for infection treatment/control, we have prioritised this common infectious complication (SBP) and a common consequence of the blood volume imbalance in cirrhosis (renal dysfunction).

Our secondary outcomes include non-SBP infections, admission to intensive care and adverse events.

If albumin is effective in preventing SBP, it may also be effective against other infections. Due to its multiple functions in the body, albumin may provide protection against other life-threatening disease complications such as hepatic encephalopathy. In this review, all other potential complications are represented by the outcome 'admission to intensive care'. Previously reported serious adverse events in albumin therapy do include cardiac disorders and respiratory disorders (pulmonary oedema, bleeding from gastric varices) and so it is important to evaluate the therapy in context of any adverse events that may have been observed in these studies. The occurrence of complications and adverse events will be assessed after the start of the albumin treatment (after the delivery of the first dose of albumin).

### Assessment of risk of bias in individual studies

To assess the risk of bias in RCTs, we will use the Cochrane Collaborations tool.[10] The Newcastle-Ottawa Scale,[11] will be used to assess the risk of bias in observational studies. Age, sex, severity and aetiology of cirrhosis, comorbidities, abstinence and treatment/medication other than the intervention/comparator will be considered the most important confounders in this assessment. Two independent review authors (SH and CP) will make the risk of bias judgements together with a justification for each judgement (a direct quote from the study where possible) using standardised forms in a web-based, systematic review management software (DistillerSR). Disagreements will be resolved by discussion and if required by consulting a third review author (AO or JR). The assessments will be presented in figures that show the risk of bias in different risk areas at the level of individual studies and the risk of bias in different risk areas across the studies.

### Assessment of bias in conducting the systematic review

We will report any differences between the methods of this pre-specified review protocol and the methods in conducting the complete review.

### Data synthesis

#### Criteria for quantitative data synthesis

We plan to perform a formal meta-analysis where >1 study per outcome is identified and we consider the studies similar enough to produce a meaningful pooled effect.

#### Measures of treatment effect

For dichotomous data, the treatment effect will be estimated and presented as a risk ratio with 95% CIs. For time-to-event data, the effect will be estimated and the results presented as a log HR with 95% CI.

#### Unit of analysis issues

The outcomes will be analysed at the level of individual study participants.

#### Dealing with missing data

To obtain outcome data that are only partially reported (eg, where only the study abstract is available or an outcome is only reported in figure format) or are missing completely (outcome was set to be measured but was not reported on), we will contact the study investigators. Where possible, we will calculate missing SD from other statistics such as CIs or standard errors.

### Assessment of heterogeneity

We plan to present a forest plot and calculate the formal heterogeneity variance statistics $\tau^2$, $I^2$ and the Q-statistic for each of the review outcomes. We will regard heterogeneity as substantial if $\tau^2$ is greater than 0, $I^2$ is more than 30% and the p value for Q-statistic is less than 0.10. We plan to explore the potential reasons for substantial heterogeneity using meta-regression (specified below).

### Quantitative data synthesis

Statistical analyses will be carried out using Stata V.15 and RevMan. To account for the presence of heterogeneity, random effects meta-analysis will be used to summarise the average effects of albumin therapy on the defined outcomes. The results will be presented separately for patients without and with baseline SBP or non-SBP infection in forest plots with the average treatment effect and the estimates of $\tau^2$ and $I^2$.

### Subgroup analysis and investigation of heterogeneity

We plan to investigate the potential reasons for heterogeneity through random-effects meta-regression analyses. Meta-regression will be performed in case we identify ≥10 studies per explanatory variable. Given the characteristics and design of the included studies allow it, we will consider severity of cirrhosis and aetiology of cirrhosis as the variables.

### Sensitivity analysis

In case the identified studies differ greatly in terms of risk of bias, we plan to conduct sensitivity analyses to investigate the impact of excluding studies with high/unclear risk of bias on effect estimates. If the included studies report separately on patients with different cirrhosis aetiologies or different degrees of cirrhosis severity, we plan to also investigate the impact of excluding patient populations with different aetiologies or severity on the effect estimates.

### Qualitative data synthesis

We will provide a narrative study result summary for all outcomes. Study characteristics (participants, interventions, comparators, outcomes, study design) of included studies will also be presented in tables categorised by outcome and by patients with and without SBP or non-SBP infection at baseline. For any outcomes where meta-analysis will not be carried out, the results will be presented in forest plots without the pooled effect estimate. All results will be discussed in the context of the previously published systematic reviews and meta-analyses on the effects of albumin use in cirrhosis.

## Meta-bias(es)

### Assessment of reporting biases across studies

We plan to investigate the presence of reporting bias using funnel plots. Formal test for the presence of reporting bias (Egger's test) will be performed where there are ≥10 studies in the analysis.

We plan to assess selective outcome reporting bias by comparing what the study set to measure and analyse with the results that were reported. Any trial protocol that can be identified will be used to aid this assessment. The presence of risk of selective outcome reporting bias will be evaluated using the Outcome Reporting Bias in Trials classification system.[12]

### Confidence in cumulative evidence

We will assess and report the overall quality of the body of evidence for each review outcome using the Grading of Recommendations Assessment, Development and Evaluation Working Group system.[13] The study quality will be assessed by two independent review authors (SH and CP).

**Contributors** The study was conceived by SH, CP, JR and AO. SH developed the eligibility criteria, search strategy, risk of bias assessment strategy and data extraction plan with guidance from CP, JR and AO. SH wrote the manuscript, to which all authors CP, JR and AO contributed.

**Funding** This work is supported by the UK Biotechnology and Biological Sciences Research Council grant number BBSRC BB/M009513/1 to SH. Funding for DistillerSR licenses is supported by the Health Innovation Challenge fund (Wellcome Trust and Department of Health) award number 164699 to AO.

**Competing interests** None declared.

**Patient consent for publication** Not required.

**Provenance and peer review** Not commissioned; externally peer reviewed.

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
