## [Reviewer comments · BMJ Open]

ARTICLE DETAILS

TITLE (PROVISIONAL)	Effectiveness of intravenous albumin therapy to prevent spontaneous bacterial peritonitis, renal dysfunction and death in adults with cirrhosis: A protocol for a systematic review
AUTHORS	Härmälä, Suvi; Parisinos, Constantinos; Ryan, Jennifer; O'Brien, Alastair

VERSION 1 – REVIEW

REVIEWER	Sebastián Marciano Hospital Italiano de Buenos Aires, Buenos Aires, Argentina
REVIEW RETURNED	10-Oct-2018

GENERAL COMMENTS	Overall I consider the protocol is well design and the research question is important and feasible. I have only some minor observations that might improve the protocol: - In page 6 the authors say that: "Patients included in multiple studies will be reported only once". Please, clarify which methodology would you apply to do so.- In page 8, it is stated that "We will identify relevant articles by combining search terms for albumin, SBP, ascites and cirrhosis." These search would only be useful to identify articles that address the primary objectives. However, there are other objectives (secondary) like "non-SBP bacterial infections". If the authors do not apply a broader search, these outcomes would not be adequately evaluated.- In page 11, the authors explain how they will evaluate adverse events. Since the will include trials and non-intervention studies (observational cohorts) I suggest them to explain in detail when this adverse events will be evaluated. For example: after the first dose of albumin, after randomization, or in any other possible time-point.
--

REVIEWER	NOUSBAUM Jean-Baptiste Hépatogastroenterology unit University hospital La Cavale Blanche Brest, France
REVIEW RETURNED	03-Nov-2018

GENERAL COMMENTS	complications of cirrhosis and death. The protocol methodology is well described from a statistical point of view. However, there are important points to discuss: - Albumin administration can restore immune competence in patients with cirrhosis, but it has not yet been proven that intravenous albumin prevents spontaneous bacterial peritonitis (SBP). This should not be a primary outcome.
---

	- There are two distinct questions : o a) the first is the prevention of hepatorenal syndrome in patients with SBP. The available meta-analyses are consistent as described by the authors (Salerno F et al. CGH 2013 ; Kwok CS et al. Biomed Res Int 2013). Four randomized trials were conducted (Sort et al. 1999, Xue et al. 2002, Fernandez et al. 2005, Chen et al. 2009). The protocol will not provide more conclusive data. o b) the second is the prevention of circulatory dysfunction in patients undergoing large volume paracentesis. The effects of albumin infusions are discussed. The meta-analysis conducted by Bernardi et al. (Hepatology 2012) included 1225 patients enrolled in 17 randomized trials from 1988 to 2010. The meta-analysis performed by Kutting et al. (J Gastroenterol Hepatol 2017) included 21 trials from 1988 to 2016 with a total of 1277 patients. Human albumin significantly reduces circulatory dysfunction induced by paracentesis. The first meta-analysis showed that albumin significantly reduced mortality by 36% compared to other treatments, while the most recent meta-analysis could not demonstrate an improved survival, although it showed a trend towards a 22% reduction in mortality with albumin use. The results of the two meta-analyses differ in terms of the inclusion of studies. Bernardi et al. decided to exclude a study comparing albumin to mannitol, which is not indicated as a volume expander and could be harmful. The authors should focus on this question with a critical analysis of each study and the control group agent. - Intravenous albumin is not effective in reducing mortality in adults with cirrhosis without SBP (Guevara et al. J Hepatol 2012, Thevenot et al. J Hepatol 2015). Two randomized trials were conducted for this purpose, it does not seem necessary to look for other data with a high degree of evidence. The severity of circulatory dysfunction impairment in non-SBP-related infections appears to be lower than in SBP, and the beneficial effect of albumin on renal function is modest. Minor comments : - Background page 4, line 11 : Chronic liver disease is not caused by fibrosis, but by an inflammation that causes fibrosis.
--	--

VERSION 1 – AUTHOR RESPONSE

Reviewer: 1

Reviewer Name:

Sebastián Marciano

Institution and Country:

Hospital Italiano de Buenos Aires, Buenos Aires, Argentina

Please state any competing interests or state 'None declared':

None declared

Reviewer's comment: Dear Authors, I read with interest the protocol entitled "Effectiveness of intravenous albumin therapy to prevent spontaneous bacterial peritonitis, renal dysfunction and death in adults with cirrhosis: A protocol for a systematic review"

Overall I consider the protocol is well design and the research question is important and feasible. I have only some minor observations that might improve the protocol.

Author's response: We thank the reviewer for the kind comments and useful feedback.

Reviewer's comment: In page 6 the authors say that: "Patients included in multiple studies will be reported only once". Please, clarify which methodology would you apply to do so.

Author's response: We thank the reviewer for noting this unclear sentence. As we describe the methodology we intend to apply in dealing with multiple reports of the same study also later on in the context of full-text review (p. 10, rows 3-6), we have now removed this unclear sentence from page 6. In addition, we have added further clarification to the relevant description in the context of full-text review and the sentence (p.10, rows 3-6) now reads: "Multiple reports of the same study (duplicate, overlapping, or companion studies) will be collated into one and, where not possible, only the most relevant report based on our eligibility criteria will be included."

Reviewer's comment: In page 8, it is stated that "We will identify relevant articles by combining search terms for albumin, SBP, ascites and cirrhosis." These search would only be useful to identify articles that address the primary objectives. However, there are other objectives (secondary) like "non-SBP bacterial infections". If the authors do not apply a broader search, these outcomes would not be adequately evaluated.

Author's response: Our search concepts and the related search terms have been designed to capture the patient populations the review aims to summarise the albumin effects in: patients with cirrhosis who have ascites and/or infection. In the search strategy this translates to: any study that in its abstract, title or keywords includes the words (or medical subject headings) "albumin and "cirrhosis" and ("ascites" or "spontaneous bacterial peritonitis" or "SBP"). Based on our preliminary searches, in the context of cirrhosis patients with non-SBP infections are specified using the words "non-SBP infection" or "infection other than SBP" (to differentiate the studies from those with SBP patients) and so the search term "SBP" captures also the studies with non-SBP studies.

To make the reasoning behind our search strategy clearer and to avoid any confusion between baseline conditions and outcomes in relation to search terms (the reviewer is absolutely correct in noting that they partly coincide), we have rephrased the sentence about the search terms (p.8, rows 33-36) to read: "We will identify relevant articles by combining search terms for albumin and the eligible base conditions of the study participants: cirrhosis and ascites and/or infection (the search terms for SBP will capture both studies that specify the participants as having "SBP", "non-SBP infections" and "infections other than SBP")." We have also corrected row numbers and added a final line that shows how the search concepts are combined in the search strategy (p. 9, Table 1).

The reviewer's on-point request for clarity in terms of what we are searching for has encouraged us to clarify the phrasing of our study objectives and description of the types of studies and participants of interest.

Accordingly, instead of (p.5, rows 38-50):

"The objectives are:

- To assess the effectiveness of intravenous albumin therapy to prevent SBP in adults with cirrhosis (without SBP or non-SBP infection)
- To assess the effectiveness of intravenous albumin therapy to prevent renal dysfunction in adults with cirrhosis, with and without SBP or non-SBP infection
- To assess the effectiveness of intravenous albumin therapy to prevent death in adults with cirrhosis, with and without SBP or non-SBP infection"

The objectives-section now reads (p.5, rows 38-50):

"The objectives are:

- To assess the effectiveness of intravenous albumin therapy to prevent SBP in adults with cirrhosis and ascites (without SBP or non-SBP infection)
- To assess the effectiveness of intravenous albumin therapy to prevent renal dysfunction in adults with cirrhosis and ascites and/or infection (SBP or non-SBP infection)
- To assess the effectiveness of intravenous albumin therapy to prevent death in adults with cirrhosis and ascites and/or infection (SBP or non-SBP infection)"

And instead of (p.6, rows 18-25):

"We will include randomized clinical trials (RCTs) and cohort (with comparison group/s) and case-control studies that investigate the effectiveness of intravenous albumin therapy to prevent SBP in patients with cirrhosis (without SBP or non-SBP infection), or renal dysfunction and death in patients with cirrhosis, with and without SBP."

The sentence referring to the study participants under the Types of studies -subsection now reads (p.6, rows 18-25):

"We will include randomized clinical trials (RCTs) and cohort (with comparison group/s) and case-control studies that investigate the effectiveness of intravenous albumin therapy in patients with cirrhosis and ascites and/or infection (SBP or non-SBP infection)."

And, finally, instead of (p.6, rows 39-44):

"We will include studies that enrol 18+ year-old adult patients with cirrhosis (regardless of severity or aetiology by any classification), and without obvious signs of baseline bacterial infection when an infection (SBP or non-SBP infection) is a study outcome."

The subsection about the study participants now reads (p.6, rows 39-44):

"We will include studies that enrol 18+ year-old adult patients with cirrhosis (regardless of severity or aetiology by any classification) and ascites and/or infection (SBP or non-SBP infection), and 18+ year-old adult patients with cirrhosis and ascites without obvious signs of baseline bacterial infection when an infection (SBP or non-SBP infection) is a study outcome."

We are grateful for the reviewer's help in improving our manuscript.

Reviewer's comment: In page 11, the authors explain how they will evaluate adverse events. Since they will include trials and non-intervention studies (observational cohorts) I suggest them to explain in detail when these adverse events will be evaluated. For example: after the first dose of albumin, after randomization, or in any other possible time-point.

Authors' response: We thank the reviewer for raising this excellent point. We intend to evaluate the adverse events after the start of albumin treatment (after the first dose of albumin) and have now added the following sentence to clarify this (p. 11, rows 41-43): "The occurrence of complications and adverse events will be assessed after the start of the albumin treatment (after the delivery of the first dose of albumin)."

Reviewer: 2

Reviewer Name:

NOUSBAUM Jean-Baptiste

Institution and Country:

Hépatogastro-entérologie unit University hospital La Cavale Blanche

Brest, France

Please state any competing interests or state 'None declared':

None declared

Reviewer's comment:

The protocol methodology is well described from a statistical point of view.

Author's response: We thank the reviewer for the kind comment and useful feedback.

Reviewer's comment:

However, there are important points to discuss:

Albumin administration can restore immune competence in patients with cirrhosis, but it has not yet been proven that intravenous albumin prevents spontaneous bacterial peritonitis (SBP). This should not be a primary outcome.

Author's response: The reviewer is absolutely correct, there have been no clinical studies that have primarily focused on this. However, given the importance of infection in cirrhosis, laboratory evidence to support a potential role for albumin, and evidence for reduction of incidence of SBP in the treatment arm in the recent Lancet ANSWER-study (Caraceni P. et al 2018), we wish to look at the incidence of infection also in other studies where it may have been included as a secondary outcome.

Reviewer's comment:

There are two distinct questions:

a) the first is the prevention of hepatorenal syndrome in patients with SBP. The available meta-analyses are consistent as described by the authors (Salerno F et al. CGH 2013 ; Kwok CS et al. Biomed Res Int 2013). Four randomized trials were conducted (Sort et al. 1999, Xue et al. 2002, Fernandez et al. 2005, Chen et al. 2009). The protocol will not provide more conclusive data.

Author's response: The reviewer is quite correct and accurately identifies the largest RCTs that have been conducted to answer this question and we may not find any further studies to help guide us. However, for the purpose of my PhD studies and in order to write a comprehensive manuscript on the use of albumin to prevent complications of cirrhosis (including a comprehensive harm profile of the intervention), we have included this analysis.

Reviewer's comment:

b) the second is the prevention of circulatory dysfunction in patients undergoing large volume paracentesis. The effects of albumin infusions are discussed. The meta-analysis conducted by Bernardi et al. (Hepatology 2012) included 1225 patients enrolled in 17 randomized trials from 1988 to 2010. The meta-analysis performed by Kutting et al. (J Gastroenterol Hepatol 2017) included 21 trials from 1988 to 2016 with a total of 1277 patients. Human albumin significantly reduces circulatory dysfunction induced by paracentesis. The first meta-analysis showed that albumin significantly reduced mortality by 36% compared to other treatments, while the most recent meta-analysis could not demonstrate an improved survival, although it showed a trend towards a 22% reduction in mortality with albumin use. The results of the two meta-analyses differ in terms of the inclusion of studies. Bernardi et al. decided to exclude a study comparing albumin to mannitol, which is not indicated as a volume expander and could be harmful. The authors should focus on this question with a critical analysis of each study and the control group agent.

Author's response: We thank the reviewer for raising this important point and completely agree that the differing results of these 2 meta-analyses are very important with regards to clinical practice and we will discuss our results in the context of these in the review manuscript. As suggested, we have also added the following sentence to the current protocol manuscript to clarify this intention (p.14, rows 37-38): "All results will be discussed in the context of the previously published systematic reviews and meta-analyses on the effects of albumin use in cirrhosis."

Reviewer's comment: Intravenous albumin is not effective in reducing mortality in adults with cirrhosis without SBP (Guevara et al. J Hepatol 2012, Thevenot et al. J Hepatol 2015). Two randomized trials were conducted for this purpose, it does not seem necessary to look for other data with a high degree of evidence. The severity of circulatory dysfunction impairment in non-SBP-related infections appears to be lower than in SBP, and the beneficial effect of albumin on renal function is modest.

Author's response: Again, the reviewer correctly identifies the major studies in this area and it seems unlikely we will identify other significant studies. However, we feel it is important to include mortality

as an outcome to provide a comprehensive assessment of the available evidence to date.

Reviewer's comment: Minor comments: Background page 4, line 11: Chronic liver disease is not caused by fibrosis, but by an inflammation that causes fibrosis.

Author's response: We are grateful for this correction and have rephrased the sentence to read (p.4, rows 10-16): "Chronic liver disease, due to fibrosis-causing inflammation, usually progresses silently and slowly until the functioning of the liver is severely compromised and patients develop life-threatening complications such as ascites, spontaneous bacterial peritonitis (SBP) and renal dysfunction."

VERSION 2 – REVIEW

REVIEWER	Sebastián Marciano Hospital Italiano de Buenos Aires, Buenos Aires, Argentina
REVIEW RETURNED	12-Dec-2018
GENERAL COMMENTS	Thanks for taking the time to answer my observation and modifying the protocol.
REVIEWER	NOUSBAUM Jean-Baptiste Hépatogastroentérologie Unit University Hospital La Cavale Blanche Brest, France
REVIEW RETURNED	02-Dec-2018
GENERAL COMMENTS	The authors have responded positively to the comments. They may not find any further studies to answer some questions, however the purpose of this study is to conduct a comprehensive review.